# Structure of Spectral Composition and Synchronization in Human Sleep on the Whole Scalp: A Pilot Study

**DOI:** 10.3390/brainsci14101007

**Published:** 2024-10-06

**Authors:** Jesús Pastor, Paula Garrido Zabala, Lorena Vega-Zelaya

**Affiliations:** 1Clinical Neurophysiology and Instituto de Investigación Biomédica, Hospital Universitario de La Princesa, C/Diego de León 62, 28006 Madrid, Spain; lorenacarolina.vega@salud.madrid.org; 2Facultad de Ciencias de la Salud, Universidad Camilo José Cela, C/Castillo de Alarcón 49, Villafranca del Castillo, 28692 Madrid, Spain; paula.garridoz@alumno.ucjc.edu

**Keywords:** bipolar montage, coherence, fast Fourier transform, K-complex, Pearson’s correlation coefficient, polysomnography, power spectra, qEEG, sleep spindles, sleep staging

## Abstract

We used numerical methods to define the normative structure of the different stages of sleep and wake (W) in a pilot study of 19 participants without pathology (18–64 years old) using a double-banana bipolar montage. Artefact-free 120–240 s epoch lengths were visually identified and divided into 1 s windows with a 10% overlap. Differential channels were grouped into frontal, parieto-occipital, and temporal lobes. For every channel, the power spectrum (PS) was calculated via fast Fourier transform and used to compute the areas for the delta (0–4 Hz), theta (4–8 Hz), alpha (8–13 Hz), and beta (13–30 Hz) bands, which were log-transformed. Furthermore, Pearson’s correlation coefficient and coherence by bands were computed. Differences in logPS and synchronization from the whole scalp were observed between the sexes for specific stages. However, these differences vanished when specific lobes were considered. Considering the location and stages, the logPS and synchronization vary highly and specifically in a complex manner. Furthermore, the average spectra for every channel and stage were very well defined, with phase-specific features (e.g., the sigma band during N2 and N3, or the occipital alpha component during wakefulness), although the slow alpha component (8.0–8.5 Hz) persisted during NREM and REM sleep. The average spectra were symmetric between hemispheres. The properties of K-complexes and the sigma band (mainly due to sleep spindles—SSs) were deeply analyzed during the NREM N2 stage. The properties of the sigma band are directly related to the density of SSs. The average frequency of SSs in the frontal lobe was lower than that in the occipital lobe. In approximately 30% of the participants, SSs showed bimodal components in the anterior regions. qEEG can be easily and reliably used to study sleep in healthy participants and patients.

## 1. Introduction

Sleep in humans is a behavioral state that alternates with waking, relative to which it is characterized by the attenuation of motor output, an increased threshold to sensory input, typical changes in central and peripheral physiology, stereotypic postures, relatively easy reversibility (distinguishing it from a coma), and diminished conscious awareness, although some kind of consciousness can be obtained during the rapid eye movement state (REM), in which dreaming takes place [1,2].

However, sleep is not a uniform state. In contrast, it is composed of different well-coordinated stages and is regulated by different brain structures located at the brainstem, diencephalon, thalamus, and cerebral cortex. In fact, this is an active state promoted by complex neural networks that are finely tuned. Normal human sleep comprises two distinct states: nonrapid eye movement (NREM) and REM sleep. In turn, NREM sleep is composed of three different stages: stage 1 NREM (N1), stage 2 NREM (N2), and stage 3 NREM (N3). Overall, NREM sleep accounts for 75–80% of sleep time. N1 sleep comprises 3–8% of sleep time. This phase occurs most frequently in the transition from wakefulness to sleep stages or following arousals. In stage N1 sleep, alpha activity diminishes, and a low-voltage, mixed-frequency pattern appears. Vertex sharp waves (VSW, 50–200 ms) are prominent towards the end of this phase. The N2 stage comprises 45–55% of the total sleep time. The typical findings include sleep spindles (SSs) and K-complexes (KCs). An SS is a 12–14 Hz waveform lasting at least 0.5 s, having a waxing and waning appearance with a main location generalized with a maximum at fronto-central regions, and it is generated within thalamic reticular neurons that produce inhibitory sequences on thalamocortical cells [3,4]. A KC is a waveform with two components (a negative wave followed by a positive wave), both of which last more than 0.5 s and are generated intracortically [5]. KCs are generalized with a maximum at the vertex [6]. Delta waves (1–4 Hz) in the EEG may first appear at N2 sleep but in small amounts. Stage N3 accounts for 15–20% of the total sleep time and constitutes slow-wave sleep, characterized by moderate to large amounts of high-amplitude, slow-wave activity. Finally, REM sleep accounts for 20–25% of sleep time. EEG tracings are characterized by low-voltage, mixed-frequency activity with slow alpha (defined as 1–2 Hz slower than wake alpha) and theta waves. REM sleep can be divided into tonic (desynchronized EEG, atonia, and the suppression of reflexes) and phasic sleep, characterized by rapid eye movements in all directions [5,7]. Moreover, transient changes in heart rate, tongue movements, the myoclonic twitching of the chin and limb muscles, and penis and clitoris erection are observed. Sawtooth waves, with a frequency in the theta range, involve the appearance of teeth on the cutting edge of a saw blade and often occur in conjunction with rapid eye movements.

The transitions from wakefulness to NREM sleep and from NREM to REM sleep are regulated by the brainstem, hypothalamus, and diencephalic structures [1,2,5,7,8], and although brainstem nuclei are involved in eye-ball movements and muscle tone, it is important to note that the latter can be considered epiphenomena and do not regulate the state of sleep.

Polysomnography (PSG) is the gold standard method for evaluating sleep in humans [9]. The minimum number of EEG channels recommended by the American Academy of Sleep Medicine is three, i.e., F4-M1, C4-M1, and O2-M1, although backup electrodes should be placed at F3-M2, C3-M2, and O1-M2 (see Appendix A, Figure A1). Other acceptable montages should be Fz-Cz and Cz-Oz with backup electrodes at Fpz, C3, O1, and M1. All the electrodes were placed according to the 10–20 international system. Additional electrodes are used to identify eye movements (electro-oculogram, EOG) and are placed approximately 1 cm from the edge of the eye and chin electrodes to record electromyography (EMG) [10]. Accessory electrodes help to stage sleep, but this is likely because the interpretation of EOG and EMG signals is simpler than that of EEG recordings collected from 19 channels.

In the literature, numerous papers have described numerical methods of diverse difficulty in describing partial aspects of sleep, such as KCs or SSs [11,12,13], transitions between phases [14,15,16], or those featuring pathologies [17,18], but few of them have been devoted to the sleep stage [19,20]. To the best of our knowledge, no comprehensive description of frequency composition, synchronization features, and typical waveform properties in a whole-scalp approach has been provided. Similar papers describing the spectral power structure in healthy humans date back more than 40 years [21,22].

In this work, we assessed two complementary goals: (i) the numerical characterization of the wake and sleep stages across the whole scalp, including the power spectra structure and synchronization, and (ii) revisiting the properties of KCs and SSs, in both cases using the most common montage used in clinical practice (i.e., bipolar double banana). Therefore, our goal is not to test the suitability of staging sleep with a new method but rather to describe the numerical properties of physiological sleep and assess the discriminability of spectral and synchronization properties across all the sleep phases. A positive and clear definition of physiological sleep is needed for subsequent studies on pathologies; therefore, the properties described in this work may help in the identification of pathologies such as RBD or the presence of abnormal sleep waveforms in epileptic patients. Furthermore, and although it is not the objective of the present work, this method can help in an easy and efficient staging of sleep, even without the need for additional channels.

The abbreviations used are listed at the end of the manuscript.

## 2. Materials and Methods

### 2.1. Participants

In this study, we retrospectively analyzed the recordings obtained from 9 men (33.4 ± 3.9 years old) and 10 women (31.3 ± 5.1 years old) referred for video-electroencephalography (vEEG) at the National Reference Unit for the Treatment of Refractory Epilepsy, University Hospital La Princesa (Spain), from 2018 to 2023. From a total of 494 patients evaluated, we selected those participants whose results were labeled as physiological recordings. These participants, without organic pathology, were referred for the assessment of possible paroxysmal events, i.e., epilepsy, nonepileptic psychogenic seizures, or cardiovascular events [23]. No epilepsy, encephalopathy, focal slowing, or any other anomalous waveforms were observed. Therefore, the resting state, active awake state, and all the sleep stages presented physiological properties. The clinical procedure was approved by the medical ethical review board of the Hospital Universitario de La Princesa (ref 5548) and was deemed “care as usual”. Under these circumstances, written informed consent was not needed.

### 2.2. EEG Recording and Sleep Staging

vEEG was performed via a 64-channel digital VEEG system (EMU64, NeuroWorks. XLTEK^®^, Oakville, ON, Canada) with 19 scalp stainless steel electrodes fixed with collodion according to the 10–20 international system; electrocardiography (ECG) and simultaneous video images were recorded continuously for 24 h. If needed, one or two EMGs in the limbs/shoulders were also added. Electrodes F7 and F8 were used for EOG recording and were referred to A1 for the sleep stage. No chin EMG was used.

The recordings were performed at a 512 Hz sampling rate with a 0.5–70 Hz bandwidth for EEG and EOG, with a 50 Hz notch on. The EMG bandwidth was 1.5–200 Hz, notch on, and the ECG bandwidth was 1.5–30 Hz, notch on. The impedance for EEG was under 25 kΩ. Artifact-free periods of at least 4 min were selected for analysis, except for the N1 stage, where the minimum period of analysis was 2 min.

Sleep staging was performed according to the following criteria [1,2,5,7,10]:

Wake (W): Physiological eye-closed resting-state EEG (rsEEG; [24]) with antero-posterior gradient and occipital waxing and waning alpha rhythm.

N1 stage: Alpha activity diminishes, and a low-voltage, mixed-frequency pattern appears. Vertex sharp waves (50–200 ms) are prominent towards the end of this phase.

N2 stage: No antero-posterior gradient, moderate amplitude mixed frequencies with the presence of SSs described as a 12–14 Hz waveform (the frequency band between 10 and 16 Hz is denoted as the sigma band) lasting at least 0.5 s, having a waxing and waning appearance with a main location generalized with a maximum at central derivations [10], and KCs, a waveform with two components (a negative wave followed by a positive one), both lasting more than 0.5 s, generalized through the scalp with a maximum at the forehead.

N3 stage: This stage involves slow-wave sleep and is characterized by moderate to large amounts of high-amplitude, slow-wave activity at 1 Hz.

REM sleep stage: Tracings are characterized by low-voltage, mixed-frequency activity with slow alpha (defined as 1–2 Hz slower than wake alpha) and theta waves. REM sleep can be divided into tonic (desynchronized EEG, atonia, and the suppression of reflexes) and phasic (characterized by rapid eye movements in all directions) phases. Moreover, transient changes in heart rate are observed. Sawtooth waves, with a frequency in the theta range, involve the appearance of teeth on the cutting edge of a saw blade.

### 2.3. Numerical Analysis

The numerical method used for qEEG has been previously described in detail [24,25,26]. Briefly, different lengths of raw records were exported from the EEG device to the ASCII file. Artifacts were excluded by the export of several artifact-free chunks, which were later combined for analysis. Although the raw recordings were digitized at 512 Hz, we downsampled them to 256 Hz. The exported files were digitally filtered by a sixth-order Butterworth digital filter between 0.5 and 30 Hz.

A differential EEG double-banana montage was reconstructed. The topographic placement of channels was defined on the scalp as the midpoint between the electrode pairs defining the channel, e.g., the Fp1–F3 channel would be placed at the midpoint of the geodesic between the Fp1 and F3 electrodes.

All the recordings were divided into 1 s moving windows with 10% overlap. The total length used during the fast Fourier transform (FFT) is directly related to the frequency precision in the power spectrum (PS). Overlap was used to minimize the effect produced by windowing. These features give rise to a maximum frequency sensitivity of 0.5 Hz.

For each window and frequency, we computed the discrete FFT of the voltage obtained from every channel to obtain the PS (in µV^2^/Hz). We also computed Shannon’s spectral entropy (SSE).

We used the classical segmentation of EEG bands (in Hz), delta (δ): 0.5–4, theta (θ): 4–8, alpha (α): 8–13, and beta (β): 13–30. Special attention was given to the so-called sigma band (σ): 10–15.

The different EEG bands are rooted in different neural systems; therefore, the synchronization of different bands can offer specific information. Linear synchronization in the time domain was assessed by Pearson’s correlation coefficient (ρ). A very useful method to assess specific band synchronization in the frequency domain is coherence (Coh) [27,28].

The mean values of all the windows for ρ and Coh were computed, and the mean correlation and coherence matrices were obtained. The mean values of synchronization for hemispheres and lobes were computed as the average of all the pairs ($N_{pairs}$) of channels ($N_{ch}$) according to the expression $N_{pairs}=\frac{N_{ch}\left(N_{ch}-1\right)}{2};N_{ch}=3\;for\;lobes\;and\;N_{ch}=9$, for hemispheres.

The spectral and synchronization variables were topographically grouped into hemispheres and lobes. In the case of the left hemisphere (shown as an example), we grouped the following topographical regions:

Frontal lobe $F=\left\{\frac{\left({Fp}_{1}-F_{3})+(F_{3}-C_{3})+({Fp}_{1}-F_{7}\right)}{3}\right\}$

Parieto-occipital lobe $PO=\left\{\frac{(C_{3}-P_{3})+(P_{3}-O_{1})+\left(T_{5}-O_{1}\right)}{3}\right\}$

Temporal lobe $T=\left\{\frac{\left(F_{7}-T_{3}\right)+\left(T_{3}-T_{5}\right)+\left(T_{5}-O_{1}\right)}{3}\right\}$.

And for the whole left hemisphere, we used the expression $$H=\left\{\frac{\left({Fp}_{1}-F_{3}\right)+\left(F_{3}-C_{3}\right)+\left({Fp}_{1}-F_{7}\right)+\left(C_{3}-P_{3}\right)+\left(P_{3}-O_{1}\right)+\left(T_{5}-O_{1}\right)+\left(F_{7}-T_{3}\right)+\left(T_{3}-T_{5}\right)+\left(T_{5}-O_{1}\right)}{9}\right\}$$

Channels from the right hemisphere were grouped accordingly.

A numerical analysis of the EEG recordings was performed with the custom-made MATLAB^®^ software R2020 (MathWorks, Natick, MA, USA).

### 2.4. Analysis of Typical Waveforms of Sleep

The features of a transient waveform are the number of phases, amplitude, duration, and scalp location. These properties are easily measured for KCs but are more difficult for SSs because the start and end are difficult to identify, as is the amplitude.

To define the properties of sleep waveforms, we visually identified, by a clinical neurophysiologist with more than 20 years of experience, a minimum of 25 (sometimes up to 35) well-defined instances during the characteristic stages of appearance. We exported periods of 12 s centered at the maximum amplitude of the KCs. We then averaged all the elements marked to obtain an average KC (aKC; see Appendix A, Figure A2A). We measured the amplitude ($V_{i},i=1,2$; in µV) and duration ($d_{i},i=1,2;\;inms)$ of both phases (see Appendix A, Figure A3) of this aKC. In the case of SSs, instead of measuring times and amplitudes on an averaged SS, which could be difficult because of the absence of a clearly defined and constant amplitude and temporal limits, we computed the average PS (aPS) of 30 instances (5 s in length) to identify the maximum power (mP) and peak frequency (pF) for the σ component (Figure A3B).

### 2.5. Differences between the Mean Spectra

To compare the aPSs of the channels, we normalized all the PSs to the area of the PS obtained for the Cz-Pz channel during wakefulness. Therefore, the proportion between spectra for different phases was maintained, but the absolute values were removed.

We had a total of 5 aPSs for every channel (e.g., W, N1, N2, N3, and REM). Every aPS can be considered a function; then, we have five functions $h_{i};i=W,N_{1},N_{2},N_{3},\;and\;REM$. We computed the absolute difference between all the functions for every channel (max∆h) according to the following equation:(1)$$max∆h=\sum_{i\neq j}\int_{f=0}^{30}\left|h_{i}-h_{j}\right|df;i,j=W,N_{1},N_{2},N_{3},REM$$
where the domain is the frequency $f\in \left[0,30\right]$.

### 2.6. Numerical Model of the Density of Sleep Spindles

To assess an estimation of the SS density and its relationship with the σ component, we performed a numerical simulation with a controlled density of SSs, measured as the percentage of SSs over a defined period. We exported 50 chunks (2 s each) of the N2 stage without SSs and another 50 periods, including a well-defined SS. We randomly selected different proportions of SSs, namely, 0, 25, 50, 75, and 100%. We computed the aPS for these periods of 100 s and later measured the pF and mP for the σ band at channels Fp1-F3/Fp2-F4 (frontopolar) and Fz-Cz (anterior midline).

### 2.7. Statistics

Evidently, the absolute values of the PS are quite different from patient to patient; therefore, it is not useful to compare the raw values. Therefore, we used a log transform [24,29] for the PS measures (logPS). The synchronization measures and SSE were compared in terms of their raw values.

Statistical comparisons between groups (for example, between different phases or between different anatomical locations) were performed via Student’s *t*-test or ANOVA for normally distributed data. Normality was evaluated via the Kolmogorov–Smirnov test. The Mann–Whitney rank sum test, Wilcoxon signed rank test, or Kruskal–Wallis one-way analysis of variance on ranks was used when normality failed. In the last case, the Tukey test was used for all the pairwise post hoc comparisons of the mean ranks of the treatment groups. Comparisons between the same lobes from different stages were assessed through paired Student’s *t*-tests. The SigmaStat^®^ 3.5 software (SigmaStat, Point Richmond, CA, USA) and MATLAB^®^ were used for statistical analysis.

Descriptive statistics are presented as the mean ± SEM (for Gaussian distributions) or median (Med), and the interpercentile range IP25-75 (percentile 25, percentile 75) for non-Gaussian distributions.

Quadratic nonlinear regression was performed via the least-square method, and *r* was used to study the dependence between the variables. Statistical significance was evaluated via a contrast hypothesis against the null hypothesis ρ = 0 via the following formula:(2)$$t=\frac{r\sqrt{n-2}}{\sqrt{1-r^{2}}}$$

This describes a one-tailed t-Student distribution with *n −* 2 degrees of freedom [30].

The SigmaStat^®^ 3.5 software (SigmaStat, Point Richmond, CA, USA) and MATLAB^®^ were used for statistical analysis.

The significance level was set at *p* = 0.05.

## 3. Results

### 3.1. Sleep Staging

Sleep staging was performed according to the AASM [10]. Stationary periods longer than 4 min were easily obtained for all the stages of sleep, except for the N1 stage, which sometimes lasted only 2 min. An example of staging and the different average spectra are shown in Appendix A, Figure A4.

### 3.2. Sex Dependence and Symmetry between Hemispheres

Before analyzing the changes in the variables between the sleep phases, we addressed the general properties for both sexes because it has been shown that females present higher absolute spectral densities [31]. In Table 1, we present the whole-scalp logPS, Coh (including in the same stage δ, θ, α, and β), and ρ for both sexes.

The whole-scalp average for logPS was lower for the W stage and greater for the N3 and REM stages, whereas the whole-scalp average for Coh (computed for all the bands) was lower for women at the W, N1, and N2 stages. No differences in whole-scalp averages for ρ were observed.

Despite the differences observed for global magnitudes, when we compared the same regions and stages between men and women, as shown in Figure 1, neither differences for any sleep phase nor for any frequency, SSE, or synchronization between males and females were observed.

Figure 1 shows that frontal power tends to be lower in the theta, alpha, and beta bands than in the other lobes of the same band.

Considering the absence of differences in paired lobes between males and females, we combined the values from both sexes.

We also compared the symmetry between the same lobes on both sides (right and left). All of these data are presented in Appendix B, Table A1. No differences between paired lobes were observed. Therefore, the values from the right and left hemispheres were grouped for the subsequent analysis.

### 3.3. Comparison of the Average logPS by Phase and Lobe

Figure 2 shows the power of all the bands (δ, θ, α, and β) for the different lobes (F, PO, and T) through the stages from the wake (W; Figure 2A) to the REM sleep stage (Figure 2E). We compared the differences for the same band over the entire scalp (horizontal lines) and the different configurations at the same lobe of all the bands (e.g., from δ to β in the F lobe), the statistical significance of which is indicated by pairs of asterisks. Each band is indicated in a specific color (black = δ; red = θ; blue = α, and green = β); the colored lines indicate differences between a pair of lobes of the same band, and the asterisks indicate significant differences between pairs of bands in the same lobe.

Interestingly, the differences between bands for the same stage were greater during the W (θ, α, and β) and N1 (θ and α) phases, although no differences between the PO and T lobes were observed. These differences vanished during the N2 stage, where only a difference between the F and PO lobes for the θ band was observed. The N3 sleep stage was the more homogeneous phase because all the lobes had the same power, which was obviously dominated by the δ band, although the remaining bands were equally similar between the scalp areas. The REM stage ultimately revealed a difference between the F and PO and the F and T lobes for the α band as well as a difference between the F and PO lobes for the θ band. These significant differences can be clearly understood from the power spectra shown in the right column of Appendix A
Figure A4. Figure 2F shows that the theta bands in the W and REM stages were the same but completely different in the alpha band.

Considering every phase, the pairs of significant differences followed the next order, from highest to lowest: N3 > REM > N2 > N1 = W.

Regarding the differences between the bands measured at the same stage and lobe, it is clear that the lower differences were for the T lobes, probably because the dispersion was greater in this region. For each lobe, there is a maximum of six pairs of different bands (i.e., δ/θ, δ/α,…α/β) and five stages (from W to REM); therefore, the maximum number of different pairs is 6 × 5 = 30. The differences are shown in Table 2.

Considering all the stages, the differences were greater for the F lobes, followed by PO, and finally T. For the F lobe, all the combinations, including the δ band, were different from the rest of the bands for all the stages and very similar for the PO and T lobes. The δ band was obviously the most variable across the dynamics of wake–sleep. However, the less different pair of bands was α/β, which was significant mainly for the F lobe (60% of the cases), and nothing at all for the T lobe.

### 3.4. Comparison of Synchronization by Phases and Lobes

In addition, PS changes for every stage in terms of both the power of the bands and the scalp distribution. We addressed the changes in the two linear measurements of synchronization used. In this case, synchronization was measured not only for every lobe but also for all the channels of the same hemisphere. Figure 3 shows the synchronization measured through the Coh of all the bands and lobes through the stages, considering the global coherence (Figure 3A) and the distribution across the scalp from wake (Figure 3B) to REM sleep stages (Figure 3E). We compared the differences for the same band over the entire scalp (horizontal lines) and the different configurations at the same lobe of all the bands (e.g., from δ to β in the F lobe), the statistical significance of which is indicated in pairs of colored asterisks.

Figure 3A shows that there was a difference between the N3 and W as well as N1 and REM stages for the δ, θ, and α bands (three pairs on 10 possibilities). Therefore, the global coherence was similar for W, N1, and N2, and for N2 and N3. The coherence of the β band was quite similar for all the phases. No difference between the δ and θ coherence was observed for any phase, but all the stages showed a difference between the δ/α and δ/β θ/α pairs. The β band showed the lowest global coherence for all the phases.

Next, we analyzed the structure of coherence across the scalp and for all the phases. Interestingly, the differences between bands for the same stage were greater during the N2 (all four bands) and N3 (all four bands) phases, with differences between F and PO and between T and PO but not between F and T. The REM sleep phase also presented differences for all four bands but no difference between the F and PO lobes for the α band. The W stage was the most homogeneous phase, and only differences for the F and PO as well as PO and T lobes for the beta band were observed.

Considering every phase, the pairs of significant differences followed the same order observed for logPS, which, from highest to lowest, was N3 > REM > N2 > N1 = W.

Regarding differences between the bands measured at the same stage and lobe, it is clear that the lower differences are for the T lobes, probably because the dispersion is greater in this region, as we observed for the PS (Figure 2). The differences are shown in Table 3.

Considering all the stages, the differences were greater for the F lobes, followed by PO, and finally T. For the F lobe, all the combinations, including the β band, were different from the rest of the bands for all the stages. The β band was obviously the most variable across the dynamics of wake–sleep. However, the less different pair of bands was δ/θ.

Although they share some common features, the structures of logPS and coherence clearly differ, indicating that both properties are independent across the different stages of sleep.

Coherence is a measure in the dominance of frequency, but in the dominance of time, the measure used was ρ. We analyzed its properties in Figure 4.

Figure 4A shows that the difference between phases was greater than that for coherence. In fact, ρ detected differences in 6/10 pairs of phases, instead of the 3/10 detected by coherence. We observed differences between W/N2, W/N3, N1/N2, N1/N3, N2/N3 and N3/REM; therefore, the stage with the greatest difference was N3, whereas REM differed only once, with N3.

Additionally, the correlation by phases and across the scalp was addressed. Interestingly, the differences between the bands for all the sleep stages were similar between F/PO and PO/T but not for F/T. Conversely, no difference was observed across the scalp during the W phase. Globally, sleep is characterized by a decrease in the correlation between the PO lobe and an increase in the variability in the T lobe. However, obviously, the magnitude of correlation could change for every lobe along the phases. To evaluate this possibility, we plotted the magnitude of correlation for every lobe and for all the phases (Figure 5). For the F lobe, we obtained four pairs that were significantly different: only two pairs for the PO lobe and another two pairs for the T lobe. Interestingly, the N1/N3 pair was different for all three lobes. Interestingly, the dynamics across the phases were similar for the F and T lobes (with greater variance in the last lobe) but completely different for the PO lobe.

In summary, the structure of synchronization, measured by ρ and coherence, is highly complex, changing from phase to phase and from lobe to lobe. In both cases, the maximum value of synchronization was achieved during the N3 sleep stage.

### 3.5. Mean Power Spectra across the Scalp

We computed the aPS normalized to the channel Cz-Pz at the W state, computed as a grand average from all the participants. The global structure (i.e., the relationship between bands for every channel and through channels) was completely similar between both hemispheres. Figure 6 shows that the aPSs were completely different between the phases. We can observe how the posterior dominant rhythm (peaked at 10 Hz) decreases and slows during the N1 sleep stage, with a peak at 8.0 Hz, while maintaining a 10 Hz component only in the occipital lobes. During the N2 stage, a prominent delta band appears, which is even greater for the N3 stage. During the N2 stage, the σ power reached a maximum at approximately 13 Hz, and the θ component was lower at 7.5 Hz. During the N3 phase, the σ power was also evident but was slower than that in N2, with a mean peak at approximately 12.0 Hz (slow SS), and the theta 7.5 Hz component practically disappeared. Finally, during REM sleep, a small alpha component at 8.0–8.5 Hz (slow alpha) can be observed with the absence of the σ band.

We computed the maximum difference (maxΔh) between pairs of normalized PSs (see Equation (1)). In this way, we could identify the region of the scalp where the difference was higher and lower for all the sleep and W phases (Figure 7).

We did not find differences between the hemispheres; although we observed a nonsignificant tendency for higher values in the left channel (*p* = 0.442; for the Mann–Whitney rank sum test), we observed great symmetry between the lobes. In fact, the linear regression between the left and right hemispheres fit very well to a straight function (r = 0.9839; *p* < 10^−4^, Student’s *t*-test, Equation (2)). We computed the average of the paired channels (e.g., (Fp1-F3 + Fp2-F4)/2)) to compare with the midline channels. From these values, we observed that there were two different regions in the scalp: one with the values of maxΔh > 50, including the channels Cz-Pz (76.74), P3-O1/P4-O2 (74.79), T3-T5/T4-T6 (71.37), and F7-T3/F8-T4 (69.61), and the rest of the scalp, with values maxΔh < 40, with the lowest values at F3-C3/F4-C4 (25.64), C3-P3 (30.69), and T5-O1/T6-O2 (35.72).

### 3.6. KC and SS Distributions across the Scalp

We analyzed the properties of the KCs and SSs across the scalp. Figure 8A shows an aKC (obtained from 32 individual KCs). The amplitude is higher in the frontal regions and lower in the occipital lobe. The amplitude from the aKC for both phases is shown in Figure 8B. The error bars are the confidence intervals (CIs); therefore, taking into account that the distribution fit well to a Gaussian distribution, the amplitude was significant when the CI did not cross the 0 level. Interestingly, (Figure 8B), the aKC did not reach statistical significance at the frontopolar regions (Fp1-F3/Fp2-F4), with the maximum amplitude at the frontal lobes (F3-C3/F4-C4, 58.0 ± 3.8 and 55.3 ± 3.8 µV, respectively, peak-to-peak) and the lower amplitude at the parieto-occipital channels (P3-O1/P4-O2, 26.2 ± 2.7 and 22.4 ± 2.1 µV, respectively, statistically significant). However, the KCs not only showed an antero-posterior gradient in the parasagittal region but also on the lateral side. In this case, the highest value was observed in the frontopolar lateral region (Fp1-F7/Fp2-F8, 65.0 ± 4.6 and 62.5 ± 3.6µV, respectively), with the lowest value in the temporo-occipital region (T5-O1/T6-O2, 7.4 ± 1.4 and 11.7 ± 1.5 µV, respectively, both statistically significant). At the midline, the gradient was not as evident, with practically the same value in the frontal as in the parietal regions (61.6 ± 5.0 for Fz-Cz and 56.0 ± 3.8 for Cz-Pz). The total duration (d1 + d2) of aKC (Figure 8C) was similar to the amplitude, with a maximum duration in the lateral frontopolar regions (Fp1-F7/Fp2-F8, 0.92 ± 0.07 and 0.95 ± 0.04 s, respectively), with a minimum duration in the temporo-occipital channels (T5-O1/T6-O2, 0.30 ± 0.04 and 0.48 ± 0.05 s, respectively). The correlations between both phases were highly significant (Student’s *t*-test), as can be observed for V1/V2 (Figure 8D; r = 0.965, *p* < 10^−4^), d1/d2 (Figure 8E; r = 0.898, *p* < 10^−4^), V1/d1 (Figure 8F; r = 0.675, *p* = 0.002), and V2/d2 (Figure 8G; r = 0.753, *p* = 0.0003).

From Figure 8D,E, we can obtain the constitutive equations describing the canonical structure of the KCs. The equation relating the amplitude of both components can be described as $V_{2}\left(V_{1}\right)=-0.21\left(\pm 0.27\right)-0.87\left(\pm 0.03\right)V_{1},\;r=0.9645$, and for the durations, $d_{2}\left(d_{1}\right)=0.52\left(\pm 0.09\right)+0.77\left(\pm 0.08\right)d_{1},\;r=0.8985$. These properties can be useful for identifying whether a waveform is a true KC.

Sleep spindles were distributed across the entire scalp (Figure 9A), but in some channels, they were difficult to distinguish from the background. Nevertheless, the presence of SSs (i.e., the σ band) can be easily identified in the PS by a well-defined σ component at approximately 13 Hz (Figure 9B). These are the reasons why we choose to analyze their distribution and properties instead of measuring durations and amplitudes. We did not observe any difference in the symmetric channels between the right and left hemispheres for pF (Figure 9C, upper row) or mP (Figure 9C, lower row). The σ component was always present in the frontopolar region (Fp1-F3/Fp2-F4, 19/19 participants), followed by the occipital region (P3-O1/P4-O2 and T5-O1/T6-O2, 18/19), while the midtemporal areas presented a lower σ band (F7-T3/F8-T4 13/19 and T3-T5/T4-T6 12/19). We addressed the average value at the N2 sleep stage for pF and mP at the F, PO, and T lobes (considering both hemispheres). We obtained a lower value (Med and 25–75 interpercentile range) for the F lobe (13.0, [12.5, 14.0] Hz and 1.05, [0.77, 1.62] µV^2^/Hz, respectively) and a higher value for the PO lobe (13.5, [13.0, 14.0] Hz and 1.07, [0.63, 3.47] µV^2^/Hz, respectively). The difference between the two regions was statistically significant for frequency (*p* = 0.020, for the Mann–Whitney rank sum test; for pF Figure 9D, upper row). No differences were observed regarding either the T lobe or the average mP (1.30, [0.83, 2.47] for F and 1.07 [0.63, 3.47] for PO, *p* = 0.566 for the Mann–Whitney rank sum test; see Figure 9D, lower row). Finally, we addressed the average pF and mP for the N2 and N3 sleep stages (Figure 9E). The frequency was statistically lower during the N3 stage than during the N2 stage (12.9 [12.7, 13.2] and 13.5 [13.3, 14.0] Hz, respectively, *p* < 0.001 for the Mann–Whitney rank sum test), and a similar result was obtained for the average mP, which was lower for N3 than for N2 (2.5 ± 0.3 and 4.2 ± 0.2 µV^2^/Hz, respectively, *p* < 0.001 for the Student’s *t*-test).

In a subset of 6/19 participants, the presence of the σ band of well-defined bimodal components during the N2 sleep stage was observed, as shown in Figure 10, where bimodal distributions at 12.5 and 14.5 Hz can be observed at several channels, e.g., Fp1-F3/Fp2-F4, F7-T3/F8-T4, or Fz-Cz. All of these bimodal σ components were obtained during the N2 sleep stage and therefore cannot be confused with the slow SS.

Finally, we addressed the relationship between the amplitude of the σ band and the density of SSs in a numerical simulation of the well-controlled percentage of SSs (Figure 11A). The average PS in the anterior regions showed an easily identifiable σ band (Figure 11B), from nearly zero (0% SSs) to a maximum when all the reconstructions included SSs (Figure 11A, lower row). The linear regression between the frontopolar (Figure 11C) and anterior midline channels (Figure 11D) was essentially one (r = 0.9964 and r = 0.9949, respectively). Therefore, there was a linear relationship between the density of SSs and the amplitude of the σ band in the anterior region.

## 4. Discussion

In this work, we revisited the spectral and synchronization properties of the wake and sleep stages for the whole scalp via double-banana bipolar recordings in participants without organic pathology. We have described these numerical properties for all the electrodes of the international 10–20 system and not for the reduced number of electrodes commonly used in PSG studies. We have shown that both the structure for band composition and synchronization are highly complex and change throughout all the phases of sleep. The numerical properties of KCs and SSs have also been described.

Usually, the PSG description of the sleep stages includes channels that do not acquire cerebral activity, such as EOG or EMG [5,10]. Although the brainstem and diencephalic structures that regulate the sleep stages project and modify the nuclei involved in muscle tone or eye movements, they do not receive feedback from the latter structures in a regulatory way. Therefore, all the sleep states, from the awake state through the different sleep stages, are the products of the neural interactions of the brainstem, thalamus, hypothalamus, and cortex; consequently, brain cortex activity is sufficient to define and differentiate all of these states [2,7,8]. This is a remarkable point because although EMGs or EOGs can be helpful for identifying different stages, they are not needed to stage sleep and wake, which are defined by EEG activity. In fact, previous studies have indicated that spectral analysis could serve as a method for sleep staging [22]. Although staging was not our goal, we have shown that the properties of the different phases of sleep and W are easy to identify and can be numerically described to implement these properties in automatic methods for sleep staging.

It has been shown that women present NREMS EEG spectra with higher absolute spectral densities than age-matched males [31,32]. Our data are not exactly the same. In fact, we did not find differences for N1 and N3 for the grand averages of logPS between men and women but we did for the stages N3 and REM. Most likely, the different configurations of the EEG recordings (referential instead of bipolar) and the widest coverage in our work, including all the IS 10–20 electrodes, could explain this discrepancy. Every EEG montage has pros and cons, and this is even more relevant for the qEEG. In fact, it was shown that a reference montage seriously biases the EEG power and coherence spectra, confounding the interpretation of the results [33]. Consequently, PS and logPS values can differ for referential and bipolar montages. Nevertheless, the global Coh for the whole band and all the scalps presented lower values for women at the W, N1, and N2 stages. No differences were observed between the sexes for ρ. However, considering not grand averages but topographical localizations, no statistically significant differences were observed between the sexes. This fact allowed us to group the values from males and females during the topographical analysis.

Differences in the average PS and synchronization throughout all the phases of sleep and across spectra have been described [22]. Our results are partially superimposed on the previous results. We have shown that slow activity increased from W to N3, with a maximum for the entire scalp. A homogeneous state is characterized by a uniform structure or composition throughout. The most inhomogeneous state was W, where the F, PO, and T lobes presented the greatest differences. However, homogeneity increased throughout sleep until the most homogeneous state was reached at N3. During the REM stage, inhomogeneity returned to the N1 level. Notably, no differences between the PO and T lobes were observed for any of the stages analyzed. There is a significant similarity between the PO and T lobes in terms of the analytical method used. The differences between the definite lobes and phases were also remarkable. These differences were greater for F lobes, where all the combinations, including the δ band, were different from the rest of the bands, followed by PO and finally T. The δ band was obviously the most variable across the dynamics of wake–sleep. In contrast, the less different pair of bands was α/β, which was significant mainly for the F lobe (60% of cases), and nothing at all for the T lobe.

Coherence between two channels is the ratio between the square of the absolute value of the cross-spectrum and the average autospectra of both; the authors of [22,24] used a measure that included only the numerator. Consequently, considering that the measurements were performed mainly between hemispheres, their measurements obviously cannot be similar to ours. Considering the band coherence for all the stages, we have shown that there is a difference between the N3 and the W, N1, and REM stages but not between N2 and any other stage. The Coh of the β band was quite similar for all the phases. No difference between the δ and θ coherence was observed for any phase, but all the stages showed a difference between the δ/α and δ/β θ/α pairs. The β band showed the lowest global coherence for all the phases. The analysis of the coherence structure for all the N2 phases revealed the greatest difference between the lobes. The most homogeneous phase was W, although differences between the F and PO and PO and T lobes were observed for the β band. The pairs of significant differences followed the same order observed for logPS. Globally considered, all the phases presented the lowest values of all the bands for the PO lobes and, systematically, for the β band. Regarding the differences between bands measured at the same stage and lobe, it is clear that the lower differences are for the T lobes, probably because the dispersion is greater in this region, biasing the statistical significance.

Although they share some common features, the structures of logPS and Coh clearly differ from phase to phase, indicating that both properties are independent across the different stages of sleep.

The variable ρ is also a measure of linear synchronization widely used in qEEG in the voltage domain [34,35]. The structure of the synchronization determined by ρ was different from that determined by means of coherence. Globally, the structure of ρ across the phases reflects coherence for the bands δ to α but not β. The scalp structure for every stage is also similar to the structure of the different bands of coherence, with a minimum value for the PO lobes and a greater dispersion at the T lobes. In terms of coherence, the W stage exhibited the most homogenous state.

The grand average PS reveals great differences between the different phases of sleep through the scalp. Obviously, the presence of the alpha band at 10 Hz was notoriously observed in the posterior regions during W. This component slowed down and moved to the anterior regions during N1. During all the phases of sleep, in both the NREM and REM sleep, a slower alpha component was easily observed at approximately 8 H [22].

The maximum amplitude of KCs has sometimes been described at the vertex [6], although the majority of authors have described the maximum amplitude at the prefrontal and frontal regions [36,37]. This distribution is in good agreement with our results, where we observed a more frontal location, with the maximum at F3-C3/F4-C4, and even more anterior at the frontopolar temporal sides (Fp1-F7/Fp2-F8). This lateral maximum for KCs was not previously described because most of the articles use a reduced number of electrodes instead of the 10–20 completed system used here. Some authors reported extremely variable morphology [38,39]; however, we obtained highly robust morphology across the participants, except for the frontopolar channels Fp1-F3/Fp2-F4, which showed great variability. In fact, we have not only observed a systematic anterior–posterior gradient for amplitude but also observed in the case of the duration with the maximum duration in the lateral frontopolar regions (nearly 1 s duration) and the shortest duration in the temporo-occipital regions (below the 0.5 s duration of the KC definition). Therefore, although KCs can be recorded through the scalp, it has a nonuniform amplitude and duration; in contrast, the anterior regions have the highest and longest waveforms, whereas in the occipital regions, the amplitude and duration are lower.

The definition of the AASM for an SS in an adult human is that of a “train of distinct waves with frequency 11–16 Hz (most commonly 12–14 Hz) with a duration ≥ 0.5 s, usually maximal in amplitude using central derivations” [10]. The reliability of visual detection between experts is generally considered to be >80% and improves further once the number of included expert scorers increases [40]. The intraspindle frequency reported is 13.3 ± 1 Hz [41,42]. We observed a Gaussian distribution with a value of 13.4 ± 1.2, which fits well with these previously published values.

SS dynamics and quantity can be measured by the mean power values in the σ band [41], and this has been the approach followed in this work. Two types of SSs (i.e., the σ band) have been described depending on the sleep phase. During the N2 stage, the faster σ component is prominent, with a pF ≥ 12.5 Hz; meanwhile, during the N3 stage, a slower component can be observed, with a pF < 12.5 Hz [43,44,45]. We observed this fact, with a σ component at 13 Hz for N2 and a slower component at approximately 12 Hz during N3. However, we are not aware of previous reports showing the presence of the bimodal components of the sigma band in the same state. We observed the presence of SSs and σ components at two frequencies during the N2 phase. We excluded the mixture of the N2 + N3 sleep stages because of the relatively small component of the delta band in the PS.

We have not addressed the different scalp locations for slow and fast SSs, although differences between the two types of SSs have been described [45,46,47]. However, we identified a pattern with two maxima in the frontopolar and occipital regions. Surprisingly, the average pF at the frontopolar regions was slower than that at the occipital lobe, which has not been previously described. This difference in pF was not associated with differences in mP. Therefore, if, as we argue, mP is an estimate of spindle density (see below), this implies that the spindle density was not modified along the fronto-occipital axis, although pF was.

Finally, the SSs are related to the macrostructure of sleep and the underlying oscillation associated with 0.02 Hz, defining the phases of the continuity and fragility of sleep [41,48,49,50]. Alterations in SS regulation can be associated with changes in microstates, such as cyclic alternating periods, which are associated with periodic leg movements or bruxism [51], or even with pathologies, such as Alzheimer’s disease [52,53]. The magnitude of the σ band can be an easy and straightforward way to estimate the density and mean frequency of SSs and could be useful in sleep medicine.

In summary, we have comprehensively and deeply described the properties of PS and synchronization structures in humans without pathology, which is the most common recording method for EEG and video-EEG recordings in clinical practice [24]. The clear definition of normative values is the first step in identifying sleep pathologies. We are aware that the number of participants is not too high; however, the results obtained, such as the PS, and the synchronization of logPS values, have been very robust. Nevertheless, more studies would be needed before the results could be generalized. Another limitation is that these participants were not healthy volunteers; in fact, some of them experienced psychological alterations, although not pathologies such as epilepsy or other structural illnesses. However, these situations, as well as the first-night effect, are more related to alterations in the hypnogram, which has not been estimated, than to the spectral or synchronization properties of the sleep states.

## 5. Conclusions

The spectral and synchronization structure is very complex and evolves throughout the different phases of sleep, although not necessarily in parallel. The average power spectra have been found to be highly specific for each of the sleep phases and could, therefore, be used (or band values) as a simple and reliable method for sleep staging. The high specificity of the spectral patterns of the different phases makes the use of accessory channels for sleep staging, such as EMG or EOG, unnecessary, as the difference between them is sufficiently clear.

We have observed that the distribution of KCs has a more peripheral distribution than assumed, with a well-defined anterior–posterior gradient. SSs have frontopolar and occipital maxima but with differences in the average frequency, which is lower in the frontal region. Another fact not previously described is the presence of bimodal spindles in a percentage of the participants. Finally, we observed that the sigma band amplitude correlates very well with the spindle density, so this method could easily be used for the study of sleep pathologies.

## Figures and Tables

**Figure 1 brainsci-14-01007-f001:**
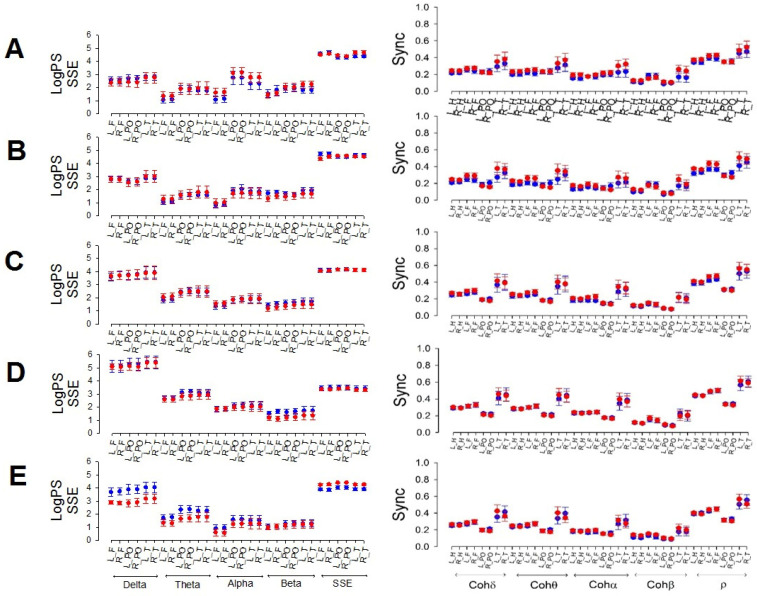
Comparison between males (red) and females (blue). The left column shows the logPS for all the bands and SSE. Right column: synchronized measurements. (**A**) Wake, (**B**) N1, (**C**) N2, (**D**) N3, and (**E**) REM. L_H = left hemisphere; R_H = right hemisphere, L_F = left frontal; R_F = right frontal; L_PO = left parieto-occipital; R_PO =right parieto-occipital; L_T = left temporal; R_T = right temporal.

**Figure 2 brainsci-14-01007-f002:**
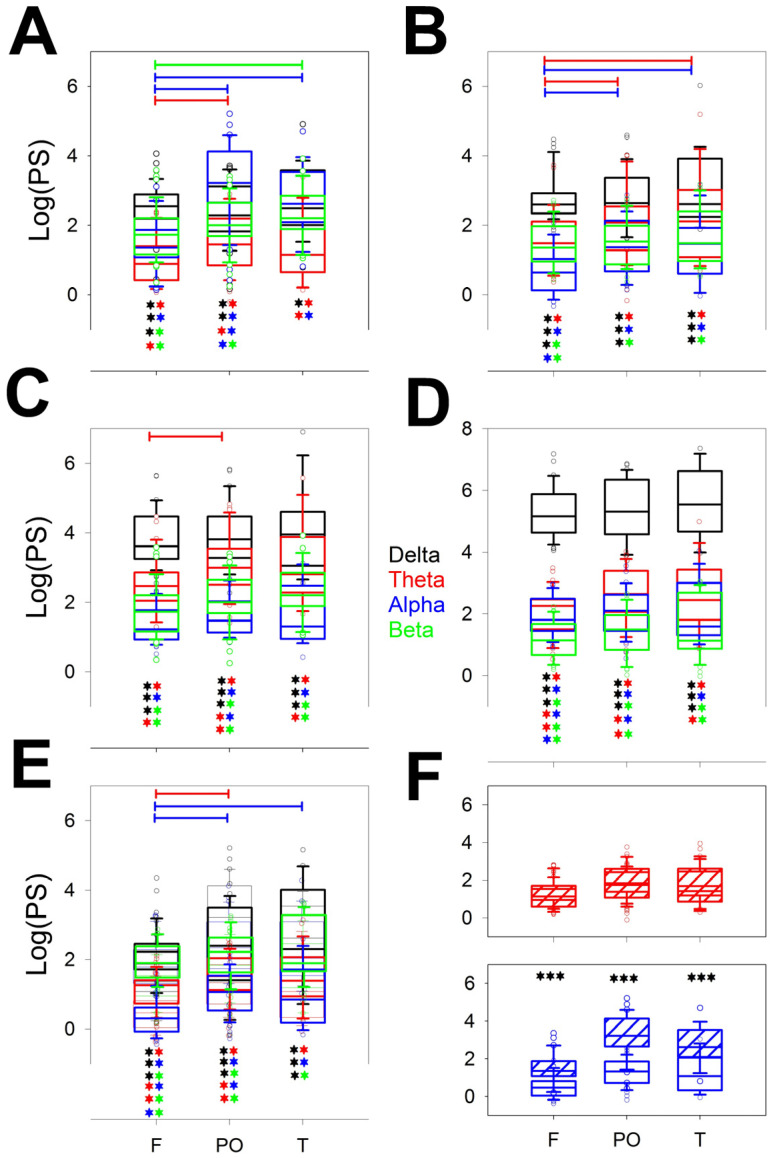
Box plot for the logPS for the different bands and lobes for the (**A**) wake stage; (**B**) N1; (**C**) N2; (**D**) N3; and (**E**) REM sleep stages. (**F**) The δ (upper row) and α (lower row) logPS values during the W stage (dashed box) and REM stage (empty box). *** *p* < 0.001 for the paired Student’s *t*-test. Delta band (δ) = black, theta band (θ) = red, alpha band (α) = blue, and beta band (β) = green. The horizontal lines indicate significant differences between pairs of lobes for the same band; the pairs of colored asterisks (representing the specific bands) indicate significant differences between the different bands of the same lobe for the Kruskal–Wallis one-way analysis of variance on ranks and Tukey post hoc tests.

**Figure 3 brainsci-14-01007-f003:**
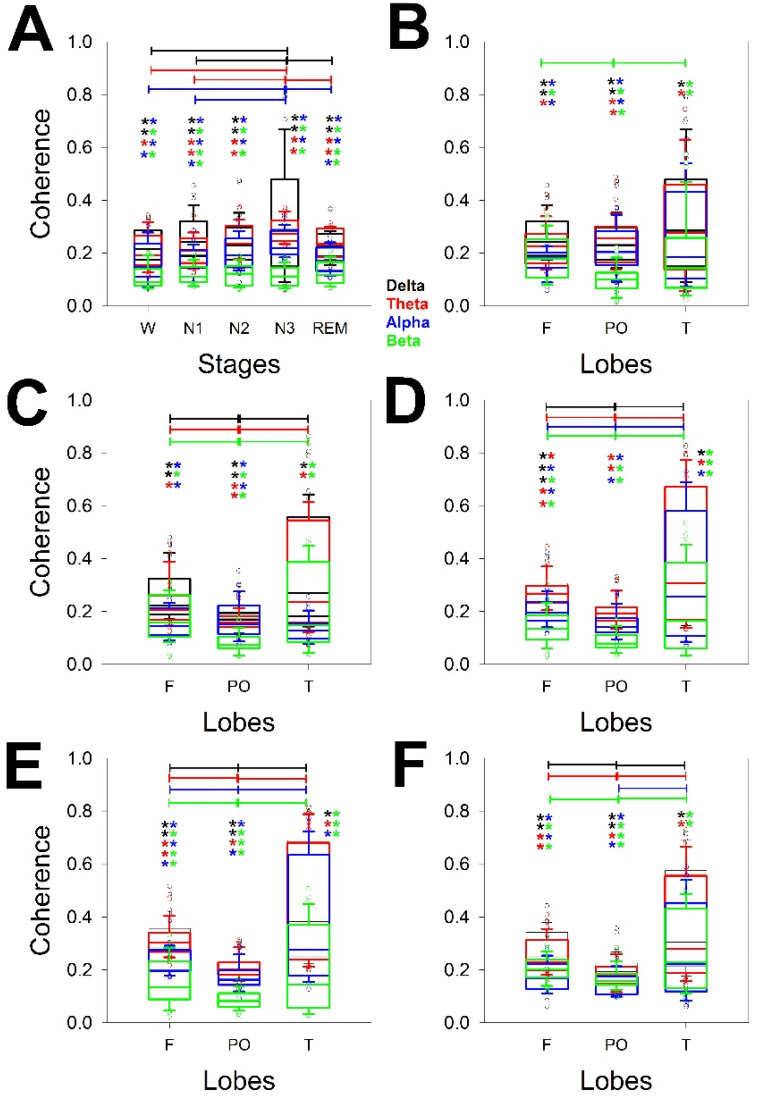
Box plot for the coherence for the different bands, whole scalp, and lobes for (**A**) the different measurements of coherence averaged for the entire scalp; (**B**) coherence during the W; (**C**) N1; (**D**) N2; (**E**) N3; and (**F**) REM sleep stages. Delta band (δ) = black, theta band (θ) = red, alpha band (α) = blue, and beta band (β) = green. The horizontal lines indicate statistical significance between pairs of phases (**A**) or lobes for the same band; the pairs of colored asterisks (representing the specific bands) indicate significant differences between the different bands of the same lobe for the Kruskal–Wallis one-way analysis of variance on ranks and Tukey post hoc tests.

**Figure 4 brainsci-14-01007-f004:**
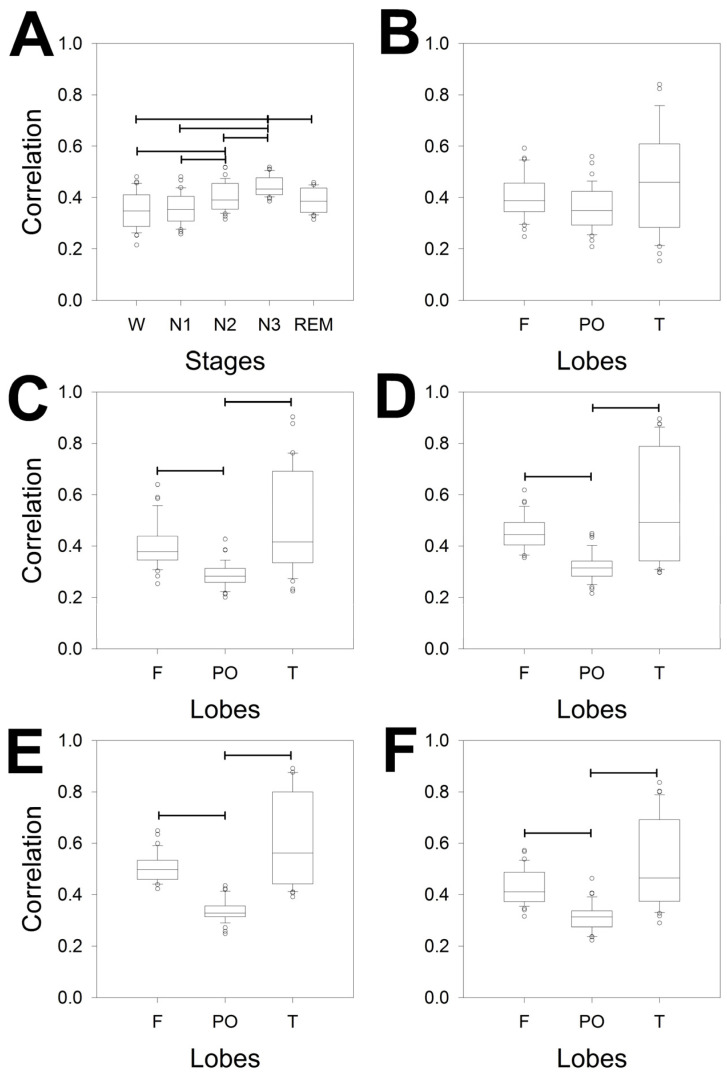
Box plot for the ρ for the different bands, whole scalp, and lobes for (**A**) the different measurements of ρ averaged for the entire scalp; (**B**) correlation during the W; (**C**) N1; (**D**) N2; (**E**) N3; and (**F**) REM sleep stages. The horizontal lines indicate significant differences between pairs of phases (**A**) or lobes for the same band for the Kruskal–Wallis one-way analysis of variance on ranks and Tukey post hoc tests.

**Figure 5 brainsci-14-01007-f005:**
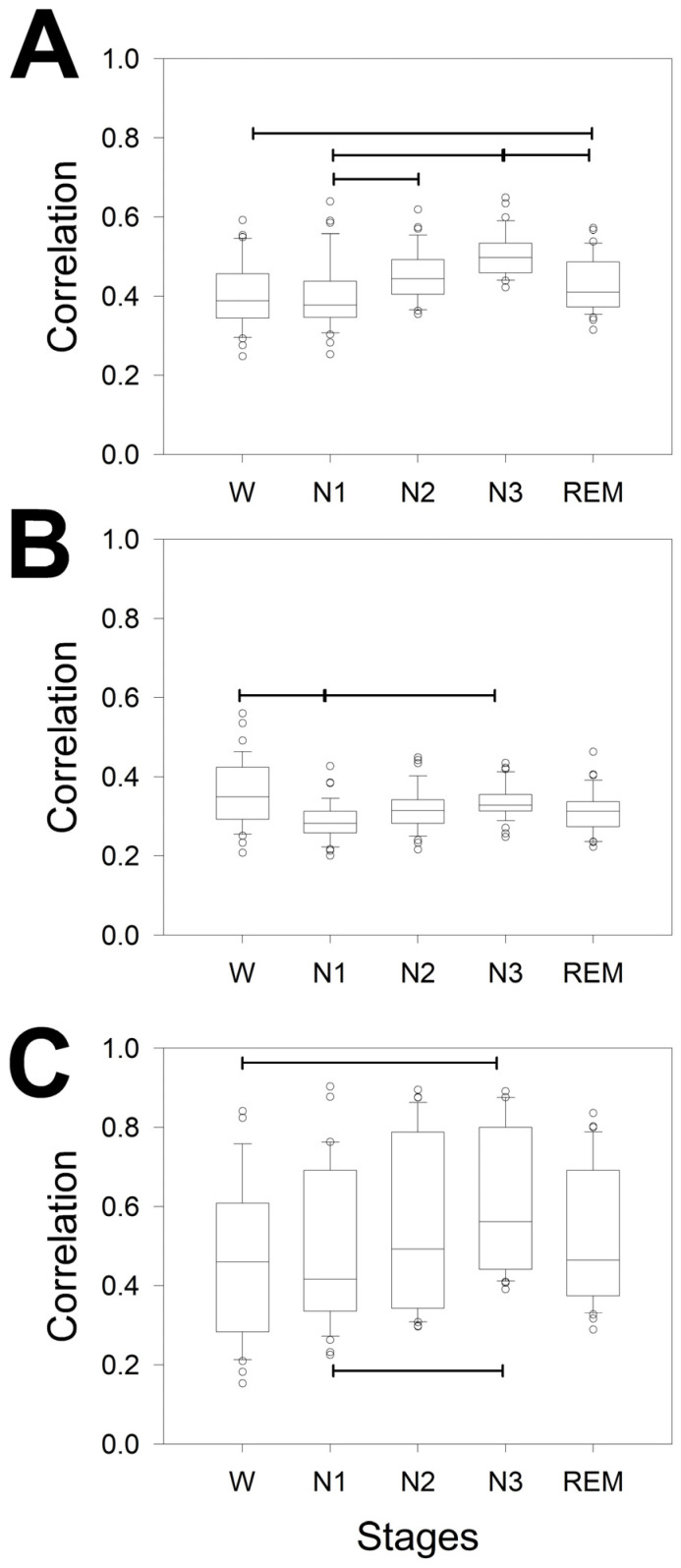
Box plot for the ρ for the different phases in specific lobes. (**A**) Frontal, (**B**) parieto-occipital, and (**C**) temporal lobes. The horizontal lines indicate significant differences between pairs of phases for the Kruskal–Wallis one-way analysis of variance on ranks and Tukey post hoc tests.

**Figure 6 brainsci-14-01007-f006:**
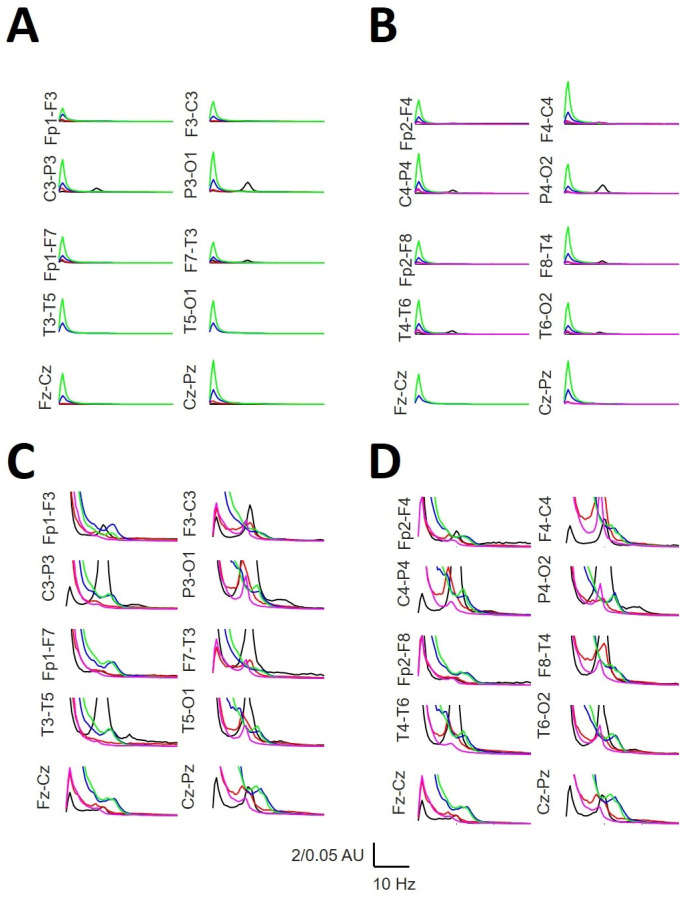
Mean averaged normalized PS. (**A**) Left hemisphere with high vertical gain; (**B**) right hemisphere with high vertical axis gain; (**C**) left hemisphere with lower vertical axis gain; (**D**) right hemisphere with high vertical gain. W = black; N1 = red; N2 = blue; N3 = green; REM = pink. AU = arbitrary units.

**Figure 7 brainsci-14-01007-f007:**
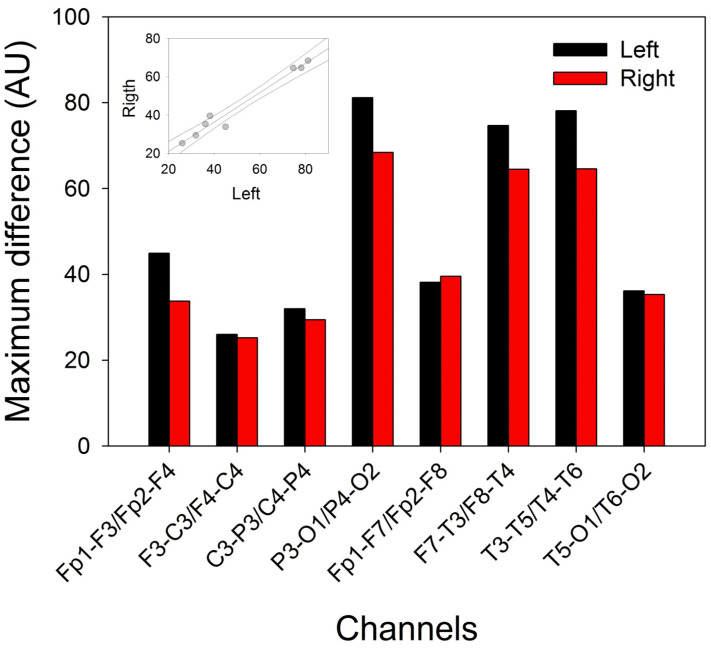
Maximum differences (maxΔh) for the average PS for every channel. Black = left hemisphere; red = right hemisphere. The inset shows the linear regression between paired channels.

**Figure 8 brainsci-14-01007-f008:**
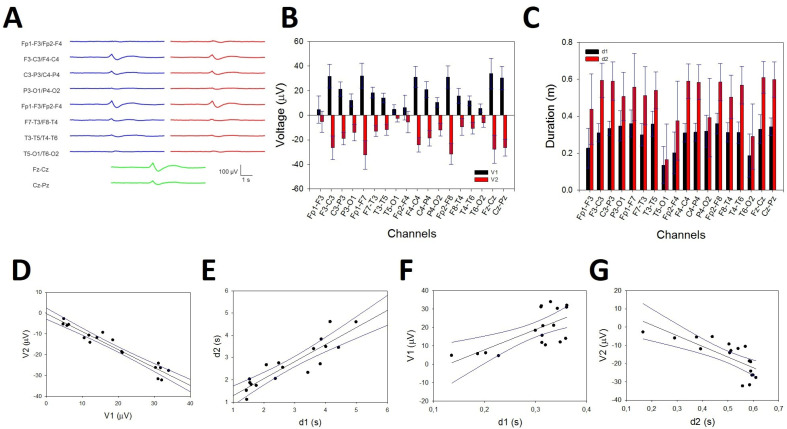
Numerical properties of KCs. (**A**) Average KCs from a patient; blue lines = left hemisphere, red lines = right hemisphere, and green lines = midline. (**B**) The bar graph of the amplitudes of the first (V1, black) and second phases (V2, red). The error bars correspond to the coefficient interval, not the SEM. (**C**) The bar graph for the duration of both phases: d1 (black) and d2 (red). The errors are the CI. (**D**) Linear regression ± 95% confidence band for the amplitude of the first and second components. (**E**) Linear regression ± 95% confidence intervals for the durations of the first and second components. (**F**) Linear regression ± 95% confidence band for the amplitude (V1) and duration (d1) of the first component. (**G**) Linear regression ± 95% confidence band for the amplitude (V2) and duration (d2) of the second component.

**Figure 9 brainsci-14-01007-f009:**
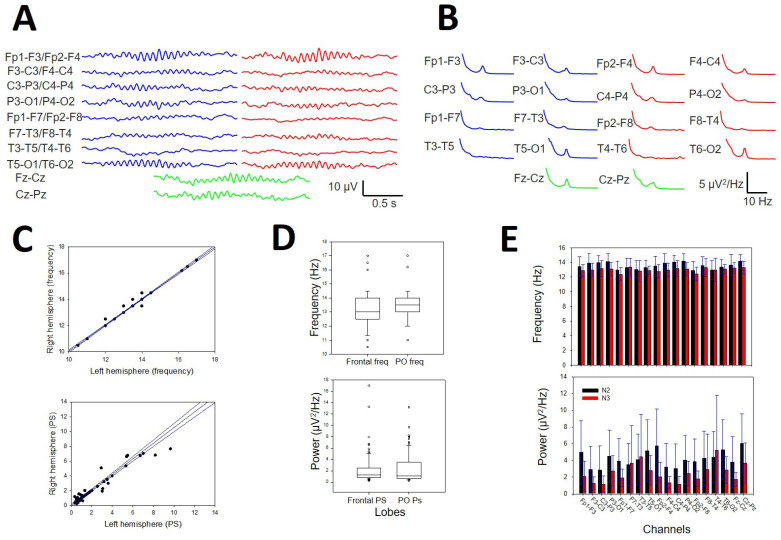
Numerical properties of SSs. (**A**) An example of SSs from a patient. (**B**) aPS for all the channels in the same patient. A prominent peak of 13 Hz (σ band) is clearly observed in 14 graphs; blue lines = left hemisphere, red lines = right hemisphere, and green lines = midline. (**C**) Upper row. Linear regression ± 95% confidence band for the average frequency of the symmetric channels in both hemispheres (r = 0.9961; *p* < 10^−4^, function y=−0.03+1.06x). Lower row. Linear regression ± 95% confidence band for the average power of the symmetric channels of both hemispheres (r = 0.9924; *p* < 10^−4^, function y=0.15+0.99x) (**D**) Upper row. Box plot for the pF at frontal and parieto-occipital lobes. Lower row. Box plot for mP frontal and parieto-occipital lobes. (**E**) The bar graph for the frequency (upper row) and the power for all the channels at the N2 and N3 sleep stages. Black = N2 stage; red = N3 stage. The error bars are the CIs.

**Figure 10 brainsci-14-01007-f010:**
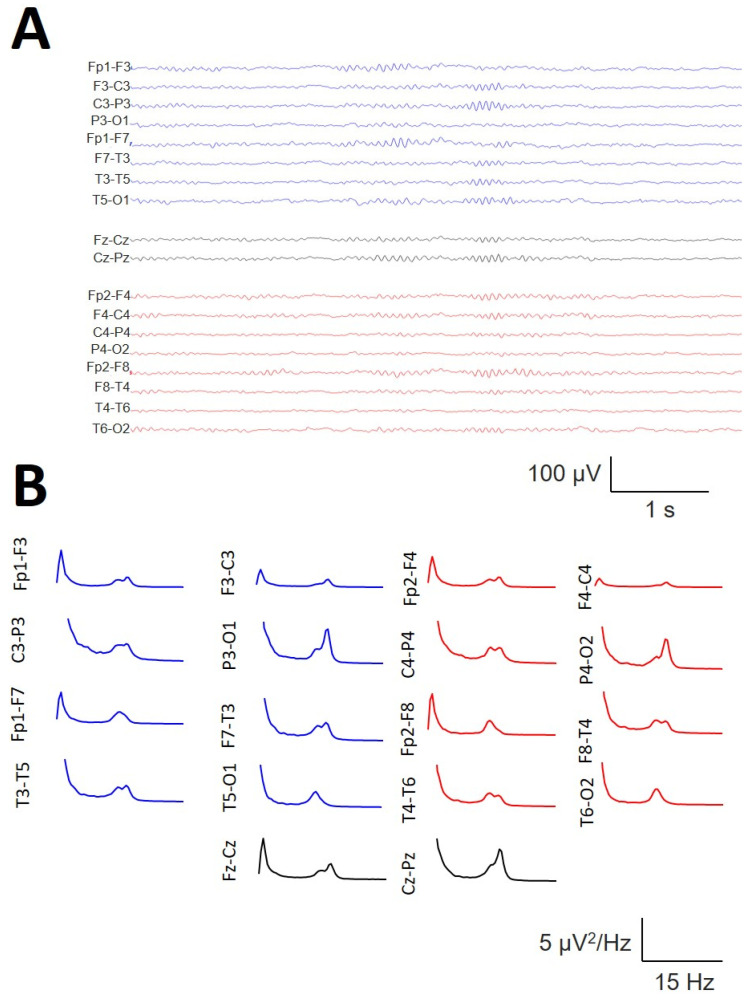
Example of a bimodal σ band in the same patient during the N2 sleep stage. (**A**) The raw trace showing the coexistence of SSs at two different frequencies (12.5 and 14.5 Hz). (**B**) The average spectra for all the channels.

**Figure 11 brainsci-14-01007-f011:**
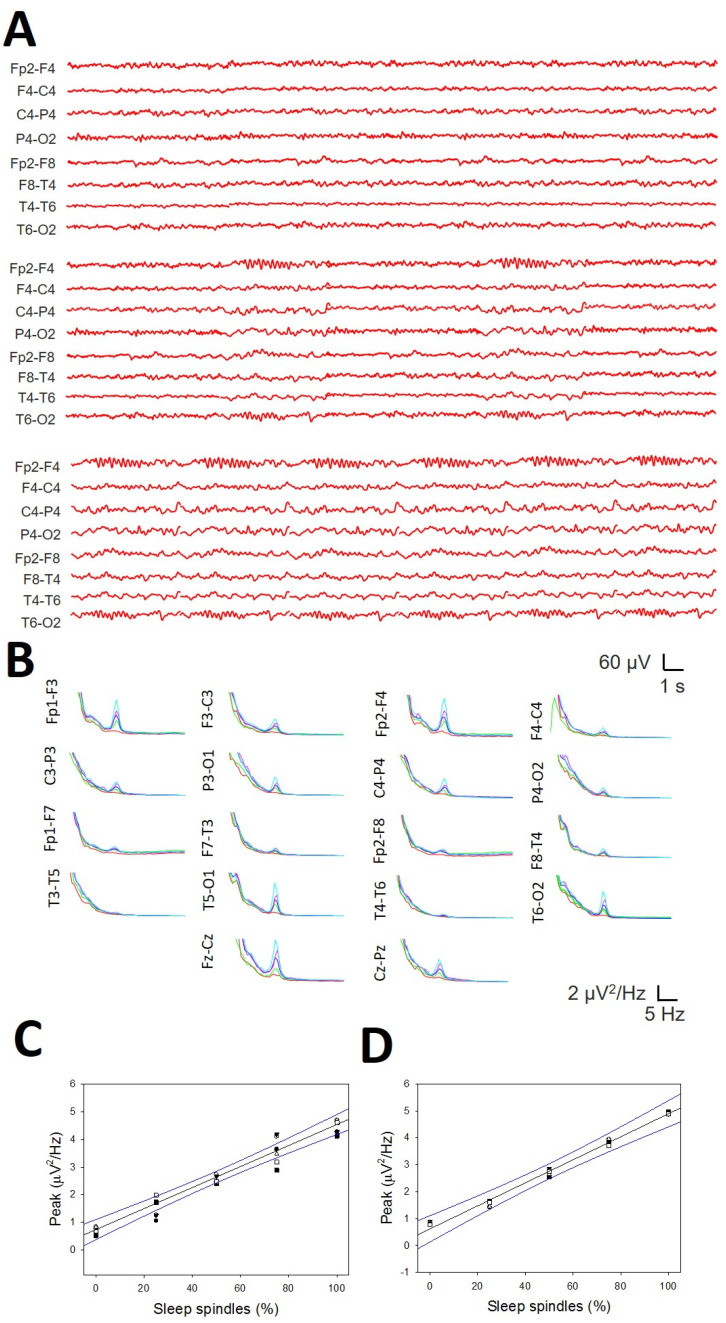
Relationship between the SS density and the amplitude of the *σ* band. (**A**) Reconstructed traces from the right hemisphere showing 0% SSs (upper row), 30% SSs (middle row), and 100% SSs (lower row). (**B**) Mean PS from one of the six simulations. Red = 0%; green = 25%; blue = 50%; pink = 75%; light blue = 100%. (**C**) Linear regression ± 95% confidence band for the amplitude of PS at the *σ* band and the percentage of SSs at the frontopolar channels (r = 0.9964; *p* < 3 *×* 10^−^^4^, function y=0.736+0.038x); (**D**) linear regression ± 95% confidence band for the amplitude of PS at the *σ* band and the percentage of SSs at the anterior midline channels (r = 0.9949; *p* < 4 *×* 10^−^^4^, function y=0.622+0.043x).

**Table 1 brainsci-14-01007-t001:** Differences between sexes for whole-scalp logPS, Coh (δ, θ, α, and β), and ρ for all stages.

Stage	logPS			Coh			ρ		
	Men	Women	*p*	Men	Women	*p*	Men	Women	*p*
W	2.18 ± 0.11	2.04 ± 0.12	0.026	0.24 ± 0.07	0.22 ± 0.06	<10^−3^	0.43 ± 0.03	0.40 ± 0.02	n.s.
N1	1.84 ± 0.12	1.88 ± 0.12	n.s	0.22 ± 0.09	0.20 ± 0.06	<0.002	0.40 ± 0.04	0.37 ± 0.02	n.s.
N2	2.32 ± 0.19	2.34 ± 0.19	n.s	0.24 ± 0.10	0.22 ± 0.09	<0.07 *	0.44 ± 0.05	0.42 ± 0.04	n.s.
N3	2.80 ± 0.31	2.99 ± 0.29	<10^−3^	0.26 ± 0.11	0.26 ± 0.11	n.s.	0.48 ± 0.05	0.47 ± 0.04	n.s.
REM	1.68 ± 0.17	2.13 ± 0.23	<10^−3^	0.23 ± 0.09	0.23 ± 0.09	n.s.	0.43 ± 0.04	0.43 ± 0.04	n.s.

n.s. = not significant; * Wilcoxon signed rank test.

**Table 2 brainsci-14-01007-t002:** Differences between all the paired bands and lobes for logPS. The data are shown as the ratio over the whole number of combinations.

Lobe	δ/θ	δ/α	δ/β	θ/α	θ/β	α/β	Σ
F	5/5	5/5	5/5	2/5	4/5	3/5	24/30
PO	5/5	5/5	4/5	4/5	3/5	1/5	22/30
T	5/5	4/5	4/5	1/5	2/5	0/5	16/30
Σ	15/15	14/15	13/15	7/15	9/15	4/15	-

**Table 3 brainsci-14-01007-t003:** Differences between all the pairs of bands and lobes in terms of coherence. The data are shown as the ratio over the number of combinations.

Lobe	δ/θ	δ/α	δ/β	θ/α	θ/β	α/β	Σ
F	1/5	5/5	5/5	5/5	3/5	1/5	20/30
PO	0/5	4/5	4/5	3/5	5/5	3/5	19/30
T	0/5	0/5	5/5	05	5/5	2/5	12/30
Σ	1/15	9/15	14/15	8/15	13/15	6/15	-

## Data Availability

The MATLAB^®^ script is available upon request from the corresponding author. The data are not publicly available due to privacy reasons.

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
