# Peer review of "Structure of Spectral Composition and Synchronization in Human Sleep on the Whole Scalp: A Pilot Study"

_brainsci, 2024, doi:10.3390/brainsci14101007_

Round 1
Reviewer 1 Report
Comments and Suggestions for Authors
Introduction
-The authors describe the appearance of PSG signals for different brain states. A figure would be very helpful in this section.
-Please explain why it is important to describe the numerical properties of physiological sleep and assess the discriminability of spectral and synchronization properties across all sleep phases. How will this benefit scientific or clinical research?
Methods
-The template paragraph describing what should be included in the Methods and Materials section needs to be removed.
Results
-Twelve figures seems excessive. Could some of these figures be moved to supplementary materials?
Discussion
-"all circadian states, from the awake state through different sleep stages, are products of the neural interactions of the brainstem, thalamus, hypothalamus and cortex" Circadian circuitry is related to, but distinct from, sleep states. Please rephrase to avoid confusion.
-The Discussion reiterates the Results. The authors should discuss how their technique is different from previous methods, with a focus on how their technique will improve outcomes or identify pathology better.
Author Response
We thank the reviewer for his comments and observations on the manuscript. Below we respond to his observations in detail.
Manuscript modifications are indicated in red.
Introduction
-The authors describe the appearance of PSG signals for different brain states. A figure would be very helpful in this section.
We have added a figure with PSG recordings (except for chin-EMG channel) as Appendix, Figure 1.
-Please explain why it is important to describe the numerical properties of physiological sleep and assess the discriminability of spectral and synchronization properties across all sleep phases. How will this benefit scientific or clinical research?
In order to clarify this point, we have added the following paragraph to the Introduction “therefore, the properties described in this work may help in the identification of pathologies such as RBD or the presence of abnormal sleep waveforms in epileptic patients. Furthermore, and although it is not the objective of the present work, this method can help in an easy and efficient staging of sleep, even without the need for additional channels.”
Methods
-The template paragraph describing what should be included in the Methods and Materials section needs to be removed.
Done. Thanks for pointing out this error.
Results
-Twelve figures seems excessive. Could some of these figures be moved to supplementary materials?
The reviewer is right that the number of figures is high. We have therefore moved Figure 1 to the Appendix. However, we feel that the rest should not be moved as they are primary data.
Discussion
-"all circadian states, from the awake state through different sleep stages, are products of the neural interactions of the brainstem, thalamus, hypothalamus and cortex" Circadian circuitry is related to, but distinct from, sleep states. Please rephrase to avoid confusion.
Thank you again for point out this mistake. We have corrected the mistake.
-The Discussion reiterates the Results. The authors should discuss how their technique is different from previous methods, with a focus on how their technique will improve outcomes or identify pathology better.
Dear reviewer, please, pay attention to these sentences taken from Discussion:
“We have described these numerical properties for all electrodes of the international 10-20 system and not for the reduced number of electrodes commonly used in PSG studies. We have shown that both the structure for band composition and synchronization are highly complex and change throughout all phases of sleep. The numerical properties of KCs and SSs have also been described”.
“we have shown that the properties of the different phases of sleep and W are easy to identify and can be numerically described to implement these properties in automatic meth-ods for sleep staging”.
“However, we are not aware of previous reports showing the presence of bimodal compo-nents of the sigma band in the same state. We observed the presence of SSs and sigma compo-nents at two frequencies during the N2 phase”.
“if, as we argue, mP is an estimate of spindle density (see below), this implies that spindle density was not modified along the fronto-occipital axis, although pF was”.
“The magnitude of the sigma band can be an easy and straightforward way to estimate the den-sity and mean frequency of SSs and could be useful in sleep medicine”.
All of these affirmations were neither previously described in such detail nor at all (e.g, the presence of bimodal SS or normative equations for KC). Therefore, we think that the manuscript describes sufficient new methods and results for the study of human sleep.

Reviewer 2 Report
Comments and Suggestions for Authors
The two complementary goals of numerically characterizing wake and sleep stages, including power spectra and synchronization, and evaluating K-complexes and sleep spindles using the bipolar double banana montage have been extensively studied, contributing little new insight to this paper.
The abstract mentioned that "In approximately 30% of patients, SSs showed bimodal components in anterior regions," but no patients were included in the dataset.
Additionally, the Materials and Methods section, particularly lines 100–115 under the Patients heading, should be removed and revised for clarity.
The abstract specifies that subjects without pathology were used, yet the paper refers to 19 subjects as patients, creating confusion. This section must be clarified.
The study was conducted with a small sample of 19 subjects, which may limit the generalizability of the results.
The focus on healthy subjects may render the study unsuitable for the Special Issue on "Diagnosis and Prediction of Neurological Diseases: Application of EEG-Based Technology."
The numerical methods presented in the paper are unclear and incomplete. They should be revised with detailed equations and proper explanations.
The visual identification of sleep waveforms should be replaced with existing methods, or the authors should propose a novel identification technique.
Moreover, all figures and equations need to be properly formatted and revised. Once these corrections are made, the results should be recalculated and presented clearly. As it stands, the paper is not ready for publication.
Comments on the Quality of English Language
Extensive editing of English language required.
Author Response
We appreciate the reviewer's effort on our manuscript and we will try to respond to all of his/her comments and observations. The modifications made to the manuscript in response to the reviewer will be in green.
The two complementary goals of numerically characterizing wake and sleep stages, including power spectra and synchronization, and evaluating K-complexes and sleep spindles using the bipolar double banana montage have been extensively studied, contributing little new insight to this paper.
Honestly, we did not find the references that the reviewer claims exist. This is probably a limitation of ours. The only articles that analyze the power spectra of sleep recordings in humans are from the last century and, of course, almost all the work done on sleep in humans uses monopolar montages and a reduced number of electrodes. ON the contrary, we use 10-20 IS with a double banana montage, rarely (if any) used in sleep research. On the other hand, we have not found any work that analyzes the numerical properties of the K complexes in the whole scalp (amplitudes, durations, relationships between phases), although there are works that perform mapping of their distribution, but always with reference montages. Therefore, we believe that our approach provides numerical data not previously published in any scientific work on sleep in humans and this increases the possibilities for scientific research in the field of sleep.
The abstract mentioned that "In approximately 30% of patients, SSs showed bimodal components in anterior regions," but no patients were included in the dataset.
We thank the reviewer for this observation, because the subjects analyzed did not present any pathology. We have corrected the error, substituting the word patients by subjects.
Additionally, the Materials and Methods section, particularly lines 100–115 under the Patients heading, should be removed and revised for clarity.
Done.
The abstract specifies that subjects without pathology were used, yet the paper refers to 19 subjects as patients, creating confusion. This section must be clarified.
The reviewer is right again. We have removed the references to patients, substituting by subjects.
The study was conducted with a small sample of 19 subjects, which may limit the generalizability of the results.
The reviewer is completely right. This is why we said in Discussion “We are aware that the number of subjects is not too high; however, the results obtained, such as the PS, as the synchronization of logPS values, have been very robust”. Nevertheless, we have added this sentence to remark this aspect: “Nevertheless, more studies would be needed before the results can be generalized”
The focus on healthy subjects may render the study unsuitable for the Special Issue on "Diagnosis and Prediction of Neurological Diseases: Application of EEG-Based Technology."
The reviewer is right again. However, the second part of the title of the Special Issue is: Application of EEG-based technology. On the other hand, this work should be considered as the numerical definition of normality in order to a further address the pathology that will be defined based on this first work.
The numerical methods presented in the paper are unclear and incomplete. They should be revised with detailed equations and proper explanations.
The reviewer is again correct in that equations are missing. This is because this is a 28-page manuscript and we did not want to make it longer. However, appropriate bibliographical references are provided, detailing the methods (see [24-26]).
The visual identification of sleep waveforms should be replaced with existing methods, or the authors should propose a novel identification technique.
Numerical methods are precise measurements that include statistical tolerance (i.e confidence intervals). They do not depend on the degree of experience of the physician. Therefore, we believe that the use of qEEG will, in the future, make visual identification of both the elements and the phases of sleep unnecessary.
Moreover, all figures and equations need to be properly formatted and revised. Once these corrections are made, the results should be recalculated and presented clearly. As it stands, the paper is not ready for publication.
We would appreciate the reviewer being more specific on this point. We do not understand what we need to revise in the figures, as they adequately summarize the data analyzed and use correct statistical tools (unless the reviewer tells us otherwise and suggests other tests). On the other hand, the fact that we modified the figures (something we do not understand), in no way implies that the results need to be recalculated, as they are robust (in the sense that, in general, they are repeatable, with small variance and important statistical significance) and correctly computed. We would appreciate the reviewer specifying what exactly is incorrect so that we can address it appropriately.
Comments on the Quality of English Language
Extensive editing of English language required.
Since our native language is not English, we have sent the manuscript to be edited by the American Journal Experts company (https://www.aje.com/es/services/translation/). I am attaching the editing certificate. I will inform the company that their work (for which they charge a good price) does not meet the minimum professional standards.

Reviewer 3 Report
Comments and Suggestions for Authors
I thank the opportunity to review the entiteled manuscript: Structure of spectral composition and synchronization in hu- 2 man sleep on the whole scalp: A pilot study, submitted to BrainSci 3191739.
I've found a very interesting study proposal, however I have several methodological concerns that I list below:
Lines 101- 115 Authors should be very careful about leaving format examples in the manuscript.
Lines 116-118, further participants information is suggested. Authors refer that "These patients, without organic pathology, were referred for the assessment of possible paroxysmal events, i.e., epilepsy, nonepileptic psychogenic seizures or cardio- vascular events" however no information about how they've got to that conclusion is provided, furthermore, they bring up with a reference from 2020, if the sample comes from a study gathered from that date, please state so.
Line 125, why do the authors say that they performed an experimental study, while it is not clear that they accomplished for experiment conditions.
Lines 127-128, I find it very difficult to have an experimental design with human participants, and no need for written informed consent? Could the authors please provide further information about this? so the readers may have clear idea about the actual conditions of the study.
Lines 133-134, What do the authors mean "if needed, one or two EMG..." what was the criteria to determine if it was needed to record EMG or not?
Line 134, Could the authors please explain why did they used dorsal frontal electrodes for EOG, instead of EOG itself? Dorsal frontal is not enough sensitive to EOG.
Lines 193-198, this is an interesting paragraph, I suggest to separate the algorithms in different lines to make it easier to read.
However, I think that the average of the topography sources need further explanations, for example, what was the time averaged for the windowing? Once I am afraid that the average involves very wide cortical areas, so resolution is lost.
Lines 199-200, This is the main idea of the manuscript, the authors should explain more details about the features of the numerical analysis, if it was custom-made this comes to be more relevant.
Line 225, Channel or stage?
Line 244, "Statistical comparisons between groups..." did the authors performed patients groups? or data from sources groups? please explain.
Figure 2. This is a very important figure, however the resolution is very bad, and it is complicated to read what the authors address at.

Author Response
We would like to thank the reviewer for his/her efforts in evaluating our manuscript. We appreciate your valuation of our work. His/her work has been inspiring and we will try to respond to his/her comments in the best possible way. Changes to the manuscript will be shown in blue.
I thank the opportunity to review the entiteled manuscript: Structure of spectral composition and synchronization in hu- 2 man sleep on the whole scalp: A pilot study, submitted to BrainSci 3191739.
I've found a very interesting study proposal, however I have several methodological concerns that I list below:
Lines 101- 115 Authors should be very careful about leaving format examples in the manuscript.
Thank you, this mistake has been corrected.
Lines 116-118, further participants information is suggested. Authors refer that "These patients, without organic pathology, were referred for the assessment of possible paroxysmal events, i.e., epilepsy, nonepileptic psychogenic seizures or cardio- vascular events" however no information about how they've got to that conclusion is provided, furthermore, they bring up with a reference from 2020, if the sample comes from a study gathered from that date, please state so.
The conclusion was obtained after the vEEG recording was analysed, as is stated at Methods “From a total of 494 patients evaluated, we selected those subjects whose results were labelled as physiological recording”. Please, pay attention that we have modified only one word.
Line 125, why do the authors say that they performed an experimental study, while it is not clear that they accomplished for experiment conditions.
The reviewer is right and we have changed the word experimental by clinical.
Lines 127-128, I find it very difficult to have an experimental design with human participants, and no need for written informed consent? Could the authors please provide further information about this? so the readers may have clear idea about the actual conditions of the study.
The reviewer is right again. The word experimental was wrong and, by replacing it with clinical, the absence of specific consent becomes much clearer, given that no action was taken on the subjects, but rather the records were analyzed off line.
Lines 133-134, What do the authors mean "if needed, one or two EMG..." what was the criteria to determine if it was needed to record EMG or not?
Some of the patients were referred for myoclonus, so the study of these patients systematically included recording of the affected muscles. None of the study subjects required these electrodes since segmental movements were not suspected.
Line 134, Could the authors please explain why did they used dorsal frontal electrodes for EOG, instead of EOG itself? Dorsal frontal is not enough sensitive to EOG.
F7 and F8 are not dorsal frontal electrodes. Instead, they are located laterally and at the temporal lobe pole. They are 1 cm further away than the EOG electrodes located at the external cantus of the eye. We have added a new figure at Appendix (Figure A1) showing the recordings obtained with the PSG montage. Pay attention to the very good recording of F7/F8 electrodes, with the presence of rolling during N1 (Figure A1A) and rapid eye movements during REM sleep (Figure A1D). Please see the next schematic figure (taken from the web and only for further clarification) to evaluate the distance between the eyeball and the F7 electrode.
Lines 193-198, this is an interesting paragraph, I suggest to separate the algorithms in different lines to make it easier to read.
We have modified the paragraph according your suggestion
However, I think that the average of the topography sources need further explanations, for example, what was the time averaged for the windowing? Once I am afraid that the average involves very wide cortical areas, so resolution is lost.
Your observations are very accurate. It is clear that grouping channels by anatomical lobes would produce a loss of spatial resolution, if we were trying to identify sources. However, our objective is not the determination of deep sources, but the global dynamics of the spectral power and synchronization of these regions. Nor do we intend to create a rigorous connectivity network (there are other much more appropriate techniques for that), but rather a representation that is sufficiently robust for clinical use.
Regarding windowing, please, pay attention to this paragraph (lines 172-175)“All recordings were divided into 1 second moving windows with 10% overlap. The total length used during the fast Fourier transform (FFT) is directly related to the frequency precision in the power spectrum (PS). Overlap was used to minimize the effect produced by windowing. These features give rise to a maximum frequency sensitivity of 0.5 Hz”.
Lines 199-200, This is the main idea of the manuscript, the authors should explain more details about the features of the numerical analysis, if it was custom-made this comes to be more relevant.
Thank you for your comment. Indeed, it is a custom-made program developed by us and implemented in MATLAB. The details are published in detail in bibliographical references 24-26. We did not want to discuse the method in more detail because, as I say, it is published and, furthermore, it is a very long work, so we considered that providing these references was enough.
Line 225, Channel or stage?
The sentence means that for each differential EEG channel there are five average power spectra (aPS), one for each of the phases studied.
Line 244, "Statistical comparisons between groups..." did the authors performed patients groups? or data from sources groups? please explain.
We have added this sentence to clarify this point“(for example, between different phases or between different anatomical locations)”.
Figure 2. This is a very important figure, however the resolution is very bad, and it is complicated to read what the authors address a
You are right, but in the final publication (we hope!) the resolution will be improved.

Round 2
Reviewer 2 Report
Comments and Suggestions for Authors
The authors have made significant progress in addressing all the raised concerns. However, the following issues still need attention:
-
Equations: In the Methods section, equations were numbered Eq. 1 and Eq. 2 but were not referenced in the main manuscript text. Additionally, the corresponding abbreviations for these equations are missing.
-
English Language: The manuscript requires significant language improvement. In many instances, it needs to be rewritten to enhance clarity and reader comprehension.
-
Wilcoxon Signed Rank Test is mentioned in the Results section, but it was not introduced in the Methods section under Statistics. Additionally, the description of post hoc test results was also missing in the Results section.
Comments on the Quality of English Language
Extensive editing of English language required.
Author Response
We thank the reviewer for his opinion on our collaboration in relation to his observations. We proceed to respond to the current ones.
The authors have made significant progress in addressing all the raised concerns. However, the following issues still need attention:
- Equations: In the Methods section, equations were numbered Eq. 1 and Eq. 2 but were not referenced in the main manuscript text. Additionally, the corresponding abbreviations for these equations are missing.
Thank you for this observation. Please, pay attention to the paragraph below Figure 6 it’s indicated “We computed the maximum difference (maxDh) between pairs of normalized PSs (see Equation (1))”. Regarding eq 2, we have included at the paragraph “In fact, the linear regression between the left and right hemispheres fit very well to a straight function (r = 0.9839; p < 10-4, Student’s t test, eq 2)”. We have added the abbreviation to the Abbreviations list.
- English Language: The manuscript requires significant language improvement. In many instances, it needs to be rewritten to enhance clarity and reader comprehension.
Please, could you be so kind to specify what paragraphs are unclear? It’s difficult and laborious to re-write completely a manuscript.
- Wilcoxon Signed Rank Test is mentioned in the Results section, but it was not introduced in the Methods section under Statistics. Additionally, the description of post hoc test results was also missing in the Results section.
The reviewer is right and we appreciate this observation. We have added the Wilcoxon at Methods/Statistics. However, we have found the expression “Tukey post hoc tests” associated with the use of the Kruskal-Wallis test.
Please, pay attention that we have improved the quality of several figures in this new version.

Reviewer 3 Report
Comments and Suggestions for Authors
I thank the authors the effort for attending my concerns. I still have a couple of minor comments:
Line 26, participants is suggested instead of subjects.
Lines 139-140, Authors argue that F7 and F8 are not dorsal frontal electrodes, according to 10-20 they are located above dorsal frontal areas, citing figure A1, I am afraid I can not read the electrodes labels on that figure due to resolution, are the very last couple of channels on D or A? EOG is commonly located around eyes commisures, if authors decided to use frontal instead, please deliver a good argument for that.
In general figures should improve before the possible publication of the manuscript.
Author Response
We appreciate the reviewer's promptness and interest in our work. Please see the PDF
